# PERSONALIZED LARGE VISION-LANGUAGE MODELS

## ABSTRACT

The personalization model has gained significant attention in image generation yet remains underexplored for large vision-language models (LVLMs). Beyond generic ones, with personalization, LVLMs handle interactive dialogues using referential concepts (*e.g.*, "Mike and Susan are talking.") instead of the generic form (*e.g.*, "a boy and a girl are talking."), making the conversation more customizable and referentially friendly. In addition, PLVM is equipped to continuously add new concepts during a dialogue without incurring additional costs, which significantly enhances the practicality. PLVM proposes `Aligner`, a pre-trained visual encoder to align referential concepts with the queried images. During the dialogues, it extracts features of reference images with these corresponding concepts and recognizes them in the queried image, enabling personalization. We note that the computational cost and parameter count of the `Aligner` are negligible within the entire framework. With comprehensive qualitative and quantitative analyses, we reveal the effectiveness and superiority of PLVM. Code is attached in the supplementary materials.

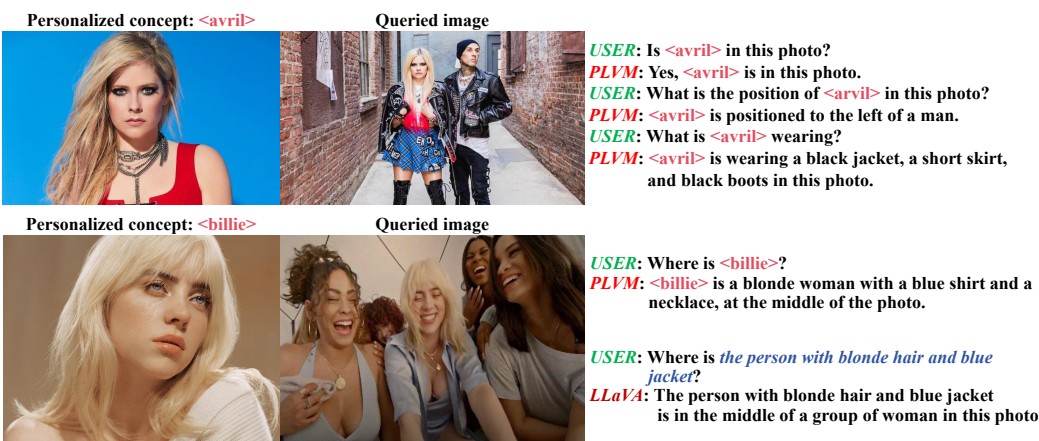

Figure 1: With personalized concepts, PLVM enhances user interaction with large vision-language models, making it easier and more intuitive.

## 1 INTRODUCTION

The personalization of AI models provides customized services to users (Gal et al.), (Ruiz et al., 2023), (Nguyen et al., 2024)), enabling the understanding of user-specific concepts (Shi et al., 2024), (Ruiz et al., 2023), (Gal et al., 2023), (Han et al., 2023), (Kumari et al., 2023), (Zhou et al., 2024), (Zhao et al., 2023), (Jiang et al., 2024). This has been extensively explored in image/video generation fields (Gal et al.), (Ruiz et al., 2023), (Zhao et al., 2023), (Jiang et al., 2024), facilitating the understanding of these concepts for personalization. For example, Dreambooth (Ruiz et al., 2023) simplifies image generation by learning a personalized concept from a few images by tuning the network parameters using a trigger word *sks* in the prompt. During the test, the image from the same identity can be generated using the fine-tuned network with the prompt with the trigger word *sks*. MotionDirector (Zhao et al., 2023) learns motion concepts from videos to conveniently render new videos with the same motions applied to different subjects. With such techniques, models perceive personalized concepts simply yet efficiently, empowering practicality.

Despite the notable advantages and successes, there remains a lack of research focused on personalization within the context of large vision-language model understanding. We argue that building personalized LVLMs for vision-language multimodal understanding can be helpful in various scenarios like dialogue systems, QA (question-answering), human-computer interaction, *etc.*. See examples in Figure 1 bottom. with personalization, a user asks a model simply by "**Where is ⟨billie⟩?**". Otherwise, the user has to describe the details of the queried subject like "**Where is the person with blonde hair and blue jacket?**" Other examples also showcase the exciting applications and practicality of personalized LVLMs.

There are two prior approaches for personalized large vision-language models – MyVLM (Alaluf et al., 2024) and YoLLaVA (Nguyen et al., 2024). MyVLM requires an additional step to verify if a personalized concept appears in an image, detecting all faces and using a classifier to identify the concept. This requires another fine-tuning process for other subjects beyond person and complicates the framework, as we aim for a unified model for both recognition and question-answering. YoLLaVA simplifies this by embedding the personalized concepts and framing recognition as a simple question: "Is ⟨sks⟩ in this photo?". However, YoLLaVA needs test-time fine-tuning and multiple images (5-10) to learn the personalized embedding. We argue that a single facial image can represent a person's identity without requiring multiple images, enhancing convenience. Furthermore, the fine-tuning process takes around 40 minutes per identity, making it inefficient for real-time applications.

This paper moves beyond the above methods and proposes personalized large vision-language models (PLVM), a simple yet efficient LVLM to understand personalized concepts like YoLLaVA (Nguyen et al., 2024). See Figure 1 for showcased abilities of PLVM. Advantageously, PLVM does not require test-time fine-tuning, significantly strengthening the practicality as incorporating new concepts will not incur additional costs. An essential design employs a pre-trained vision network (referred to as `Aligner`) to cast the given personalized concepts into online features and align them with the queried images. When prompting a concept, Aligner casts only a reference image of this concept to features, which serves as an identifier to recognize the subject of the queried image without requiring any other fine-tuning processes. We are introducing `Aligner` only conditions negligible additional cost, which will be showcased in the experiment.

The contributions of this paper are two-fold. First, we present PLVM, a personalized large vision-language model with solid abilities to recognize personalized concepts and freely incorporate new concepts without requiring additional costs. Second, PLVM achieves superior performance in a relatively simple way compared to existing methods. We also provide comprehensive experiments to showcase the effectiveness of PLVM.

## 2 RELATED WORK

**Large Vision-language Models (LVLMs).** Large language models (LLMs) (Brown, 2020; Chung et al., 2024; Thoppilan et al., 2022; Chowdhery et al., 2023; Zhang et al., 2022a; Touvron et al., 2023; Zeng et al., 2022; Chiang et al., 2023) have opened a new era of AI, showcasing their potential to handle a wide array of language-based understanding and generation tasks. To extend the capabilities of LLMs to visual understanding, the computer vision community has focused on aligning language and vision data within a shared feature space (Radford et al., 2021). Research in this area primarily follows two approaches: *internal* adaptation methods (Alayrac et al., 2022), which integrate cross-attention within LLMs for visual-language alignment, and *external* adaptation methods (Li et al., 2023; Liu et al., 2024), which involve training modules for this purpose. Consequently, vision foundation models, particularly vision transformers (Dosovitskiy et al., 2020; Liu et al., 2021; Radford et al., 2021; Tian et al., 2023; 2024b; Zhang et al., 2022b; Kirillov et al., 2023), have evolved into LVLMs (Liu et al., 2024; Tian et al., 2024a; Zhang et al., 2023; Lai et al., 2024; Qiu et al., 2024), endowing them with the capacity for language-guided visual understanding. Despite these successes, research on their personalization is scarce.

**Personalization in Image & Video Generation.** Personalization in image and video generation aims to incorporate individualized concepts into pre-trained text-to-image or text-to-video diffusion models, generating specific personalized concepts across diverse contexts. Methods for image and video personalization generally fall into two categories: test-time fine-tuning approaches (Ruiz et al., 2023; Gal et al.; Kumari et al., 2023; Han et al., 2023; Voynov et al., 2023; Liu et al., 2023), which

involve fine-tuning word embeddings (Gal et al.) or network parameters (Ruiz et al., 2023) using a limited number of examples of the personalized concept; and encoder-based methods (Shi et al., 2024; Zhou et al., 2024; Gal et al., 2023; Ye et al., 2023; Wang et al., 2024; Ruiz et al., 2024), which eliminate the need for test-time fine-tuning by encoding the personalized concepts via a pre-trained vision encoder (Radford et al., 2021; Oquab et al.). The encoded features are then integrated into diffusion model components, such as word embeddings (Shi et al., 2024; Gal et al., 2023) or network parameters (Ruiz et al., 2024; Ye et al., 2023), to facilitate the generation of personalized content. The personalization of these generative models has dramatically improved their usability. This paper aims to apply this success to LVLMs.

**Personalization in Large Vision-Language Models.** In large vision-language models, when querying about a specific subject in an image, it is typically necessary to describe the visual appearance of that subject through prompts. Personalizing LVLMs involves enabling the model to recognize specific identities, such as ⟨avril⟩ and ⟨billie⟩ in Figure 1, without requiring manual prompts for each identity. Previous methods, such as MyVLM (Alaluf et al., 2024) and YoLLaVA (Nguyen et al., 2024), employ techniques similar to textual inversion (Gal et al.) to learn the word embeddings of personalized concepts. However, these approaches require test-time fine-tuning, which is computationally intensive and costly for learning new concepts, leading to inefficiencies. Our method addresses this limitation by introducing an encoder-based approach that significantly reduces the need for fine-tuning, thereby improving efficiency.

## 3 PERSONALIZED LARGE VISION-LANGUAGE MODELS

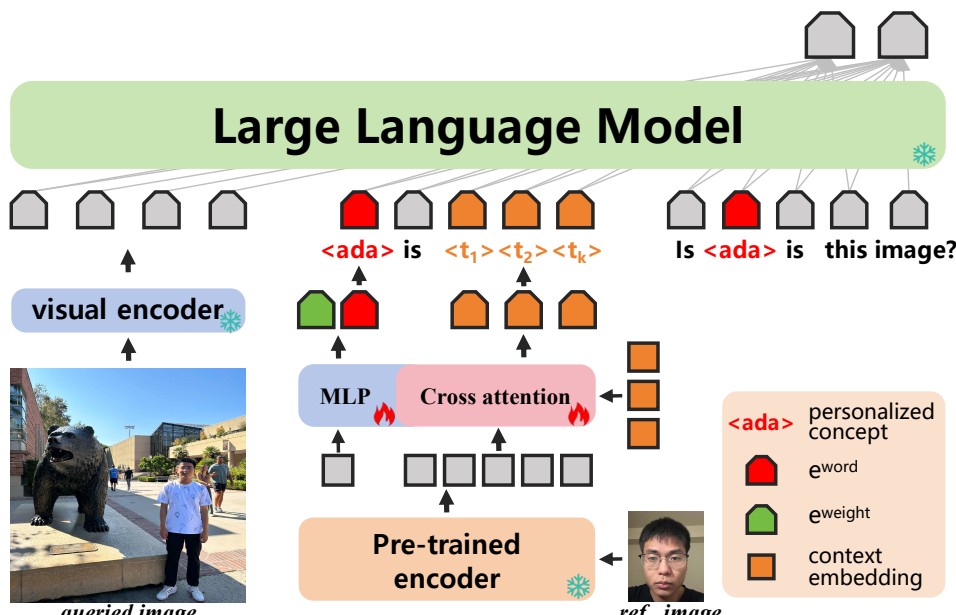

Figure 2: The overall framework of PLVM, where the large language model receives the spatial features, the concept template of a personalized reference image, text prompt, and produces the answer.

Personalized large vision-language models enhance practicality and efficiency in applications such as dialogue systems. The proposed PLVM facilitates this capability using the straightforward yet effective `Aligner` method. This section outlines the `Aligner` and then describes the training strategy and synthesis dataset designed to improve PLVM's efficiency.

Figure 2 illustrates the PLVM framework, which includes the `Aligner` for generating personalized identifiers, a large language model (based on LLaVA (Liu et al., 2024)) for visual QA, and the integration of text and image inputs.

### 3.1 ALIGNER: ONLINE ENCODING OF PERSONALIZED CONCEPTS

As previously mentioned, we use a pre-trained vision encoder to build the `Aligner`, which embeds personalized concepts online, with its output serving as an identifier for recognizing queried images.

**Architecture.** Specifically, we use the DINO-v2 (Oquab et al.) vision encoder, denoted as $E_{ref}$, to extract features from the reference image of the personalized concept, Figure 2. Given a reference image $I \in \mathbb{R}^{H \times W \times 3}$, where $H$ and $W$ are the height and width, we input the image into $E_{ref}$ to obtain visual features, represented as $z_{ref} \in \mathbb{R}^{L \times d}$, where $L = 257$, is the sequence length and $d$ is the feature dimension.

For the personalized concept $\langle sks \rangle$, we predict its word embedding ($e^{word}$) and head weight ($e^{weight}$) using $z_{ref}$. We apply 2 MLP modules, $f_{\phi_1}$ and $f_{\phi_2}$, to map the first (global) token of $z_{ref}$, generating $e^{word} = f_{\phi_1}(z_{ref}[0])$ and $e^{weight} = f_{\phi_2}(z_{ref}[0])$.

To reduce the computational cost of processing visual tokens in large language models, we introduce $k$ ($k \ll 257$) context embeddings to integrate the outputs of $E_{ref}$ (excluding the global token). These are processed by a small transformer network $\mathcal{T}_\theta$ with $k$ learnable queries $Q = q_1, q_2, \ldots, q_k$. Each query acts as a soft, learnable token. Using cross-attention, $z_{ref}$ serves as $\mathbf{K}$ and $\mathbf{V}$, while the learnable queries serve as $\mathbf{Q}$, reducing 256 tokens to 16, aligning with YoLLaVA (Nguyen et al., 2024).

As shown in Figure 2, we freeze the LLaVA and `Aligner` modules and fine-tune $f_{\phi_1}$, $f_{\phi_2}$, and $\mathcal{T}_\theta$. We also render $e^{word}$, $e^{weight}$, and the context embeddings learnable. Then, given the question $X_q$, the answer $X_a = \{x_a^1, x_a^2, \ldots, x_a^N\}$, and the query image $I_q$, we use the mask language modeling loss to compute the probability of target response $X_a$:

$$\mathcal{L}(X_a | X_q, I_q) = \frac{1}{N} \sum_{i=1}^{N} \mathcal{L}_{ce}(p_i, x_a^i), \quad (1)$$

here, $\mathcal{L}_{ce}(\cdot, \cdot)$ represents the cross-entropy loss between the predicted answer and the ground truth, $p_i = p(x_a^i | x_a^{<i}, X_q, I_q)$ is the distribution of predicting the word $i$, and $N$ denotes the sequence length of the answer.

**System prompt**. Following YoLLaVA (Nguyen et al., 2024), let $\langle sks \rangle$ represent the personalized concept and $\langle token_1 \rangle, \langle token_2 \rangle, \ldots, \langle token_k \rangle$ represent the context embedding tokens for a reference image. We inject the template prompt as "$\langle sks \rangle$ is $\langle token_1 \rangle \langle token_2 \rangle \ldots \langle token_k \rangle$" as the instruction prefix, resulting in the overall system prompt:

User: $\langle sks \rangle$ is $\langle token_1 \rangle \langle token_2 \rangle \ldots \langle token_k \rangle$ + instruction   Assistant:

For each personalized concept, we assign a unique $\langle sks \rangle$, such as "$\langle avril \rangle$" or "$\langle billie \rangle$" as shown in Figure 1. The `Aligner` and $\mathcal{T}_\theta$ transform the corresponding reference image into $k$ context embeddings. Unlike previous methods, this process works online, allowing our approach to freely integrate new concepts without additional costs like a fine-tuning process.

### 3.2 TRAINING

We observed that training solely with the objective in Equation 1 yields suboptimal results for recognition capabilities. This issue arises because, during training, the number of available question-answer pairs for task recognition is limited, whereas paired reference and query images are abundant. Consequently, the model tends to overfit the words following a "Yes" or "No" answer. For instance, if the question is "Is $\langle sks \rangle$ in this photo?", and the answer is "Yes, $\langle sks \rangle$ is in this photo.", once the model recognizes the answer is "Yes," it can easily predict the phrase "$\langle sks \rangle$ is in this photo". As a result, the most challenging recognition aspect is accurately predicting the words "Yes" or "No". To address this, we propose assigning greater weight to the loss associated with the words indicating the positive or negative response ("Yes" or "No") in the recognition QA task. Specifically, the loss function for recognition QA is:

$$\mathcal{L} = \frac{1}{W} \sum_{i=1}^{N} (w \cdot \mathbb{1}_i + (1 - \mathbb{1}_i)) \mathcal{L}_{ce}(p_i, x_a^i), \qquad (2)$$

where $w$ represents the weight associated with the loss for words indicating positive or negative answers, such as "Yes" and "No". The indicator function $\mathbb{1}_i$ is used to identify whether the word $i$ corresponds to a positive or negative response. The normalization factor, $W$ is defined as $1/(\sum_i w \cdot \mathbb{1}_i + (1 - \mathbb{1}_i))$, which adjusts the weight distribution to ensure proper scaling.

### 3.3 DATA SYNTHESIS

A significant challenge we encountered was the inability to obtain the required dataset. To overcome this issue and facilitate our method, we construct a dataset comprising paired images, specifically reference facial images and various corresponding images. For this purpose, we utilized the IP-Adapter method (Ye et al., 2023), which is pre-trained on the Stable Diffusion model (Rombach et al., 2022). Next, ChatGPT is employed to generate 200 descriptions based on the templates, such as "A person (wearing something) is (doing something) at (some place)." These prompts are then input into the IP-Adapter, conditioned on facial images from the CelebA-HQ dataset (Lee et al., 2020), to generate diverse images of the same person from the CelebA-HQ dataset. To ensure quality, we filter out low-quality images using two criteria: a CLIP cosine similarity score between the image and its corresponding prompt of less than 0.2, and a face similarity score (Deng et al., 2019) of less than 0.5. The examples of the synthetic dataset are shown in Figure 3.

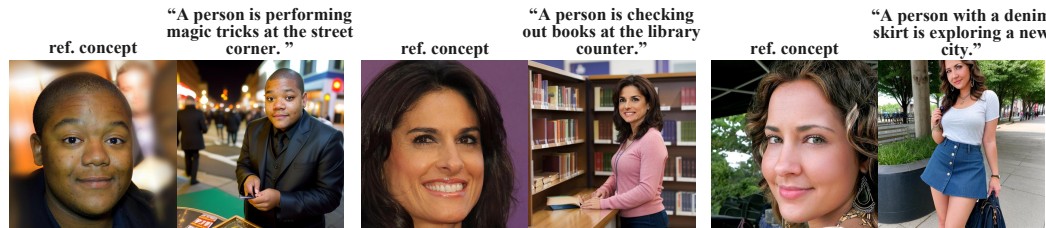

Figure 3: Examples of our synthetic data using Diffusion models (Rombach et al., 2022), which is capable of generating customized images based on a reference appearance and a given prompt.

Following the methodology outlined in (Nguyen et al., 2024), we construct a dataset for two categories of questions, *i.e.* recognition and attribute questions. For recognition questions, given a reference image depicting the face of a personalized identity, the task is to determine whether the model can identify the presence of the individual in the query image. As in (Nguyen et al., 2024), we employ a set of template questions designed for binary "Yes" or "No" answers. For attribute questions, we inquire about details such as one person's hair color, eye color, and skin tone. Consistent with (Nguyen et al., 2024), we utilize the pre-trained LLaVA model (Liu et al., 2024) to generate responses to these questions based solely on the reference image. We provide the template details in Sec. A.2.

## 4 EXPERIMENT

### 4.1 SETUPS

**Implementation details.** An MLP module consists of 3 linear layers with a hidden size of 4096 and employs the GeLU activation function to predict both $e^{word}$ and $e^{weight}$. To process the context embeddings, we adopt a cross-attention transformer architecture comprising four transformer blocks with a hidden size of 1024. During training, we allocate a portion of the samples for recognition tasks and another for detail attribute tasks, setting the default ratio for recognition to attribute at 1:1. For detail attribute-related tasks, query images are excluded from the input. The ratio of positive samples to the total is $p$, which will be analyzed in the ablation study. Unless otherwise specified, $p$ is set to 0.5 by default.

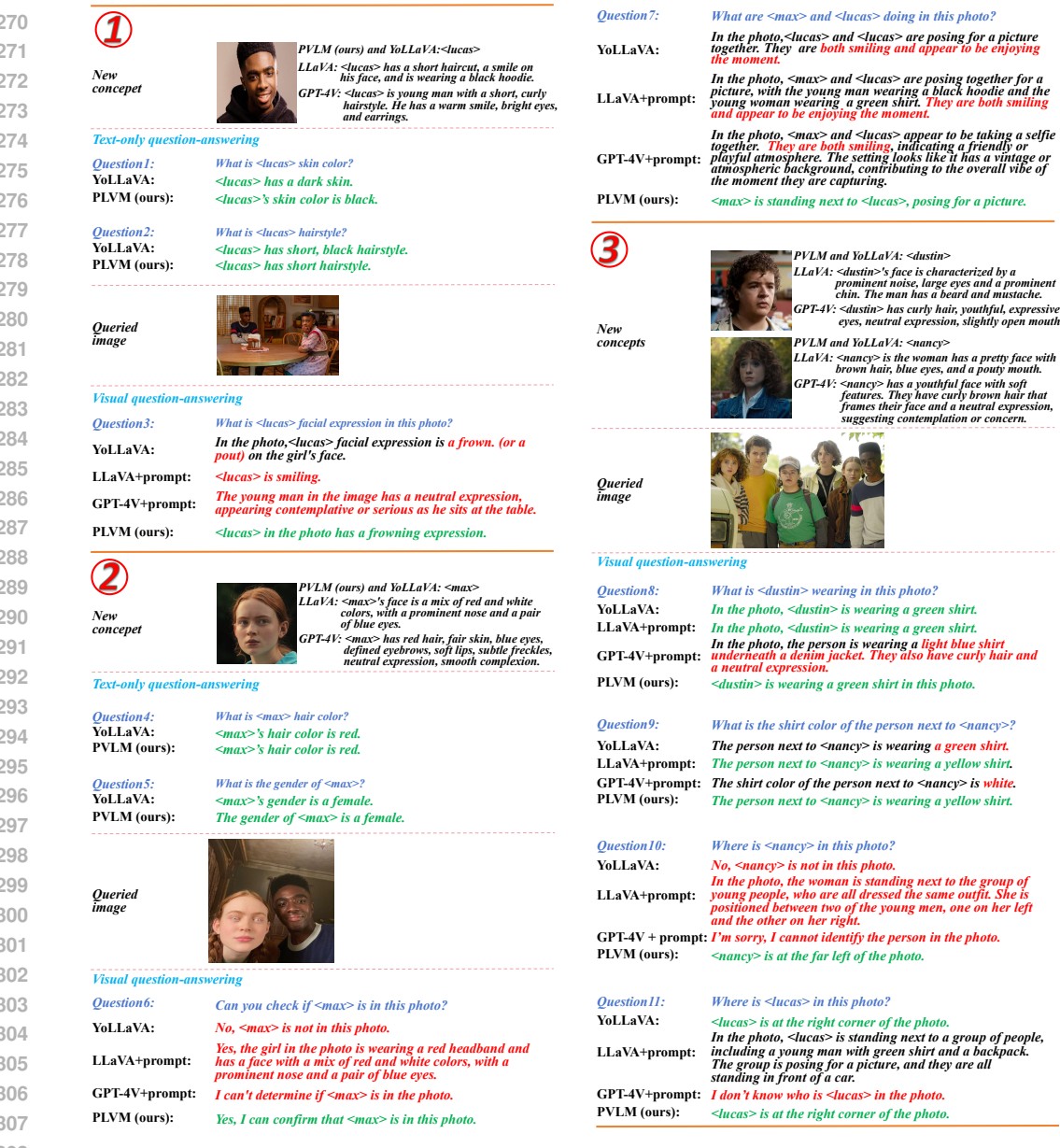

Figure 4: Qualitative results compared with YoLLaVA, LLaVA (with prompt), and GPT-4V (with prompt). PLVM requires no fine-tuning for every concept, enabling seamless incorporation of new concepts, as shown in the figure. In contrast, YoLLaVA needs ∼40 minutes of fine-tuning for each new concept to achieve personalization.

All parameters are frozen except those in the MLP and cross-attention modules. The AdamW optimizer is used with a learning rate of $1 \times 10^{-4}$ and a batch size of 2. All experiments are conducted on a single A6000 GPU.

**Dataset generation and evaluation.** As described in Section 3.1, we generate 200 prompt templates and use each image from the CelebA-HQ dataset as a reference image. Five new images are generated using randomly sampled prompts for each reference image. Low-quality images are filtered based on two criteria: a CLIP cosine similarity score below 0.2 and an ArgFace cosine similarity score below 0.5. This process results in a final dataset of 70k images.

For validation, we curate a test dataset by manually downloading images from the Internet. The dataset is organized by identity, each corresponding to a distinct concept. Each identity contains one

Table 1: The evaluation results on three types of questions, compared to other methods on the test dataset. This demonstrates that our PLVM achieves superior performance across all these tasks and showcases the effectiveness of our method in comparison to these counterparts.

| Method | Visual recognition | | | Question answering | |
|---|---|---|---|---|---|
| | Positive | Negative | Mean | Text-only | Visual |
| LLaVA-1.5 (Liu et al., 2024) (w/o prompt) | 0.0 | 100 | 50.0 | – | – |
| LLaVA-1.5 (Liu et al., 2024) (w/ prompt) | **80.3** | 61.3 | 70.8 | 71.8 | 72.3 |
| YoLLaVA (Nguyen et al., 2024) | 73.1 | 73.7 | 73.4 | 77.4 | 72.9 |
| GPT-4V (Achiam et al., 2023) | 40.5 | **97.5** | 69.0 | 47.9 | 50.5 |
| PLVM (Ours) | 78.5 | 84.7 | **81.6** | **85.9** | **77.5** |

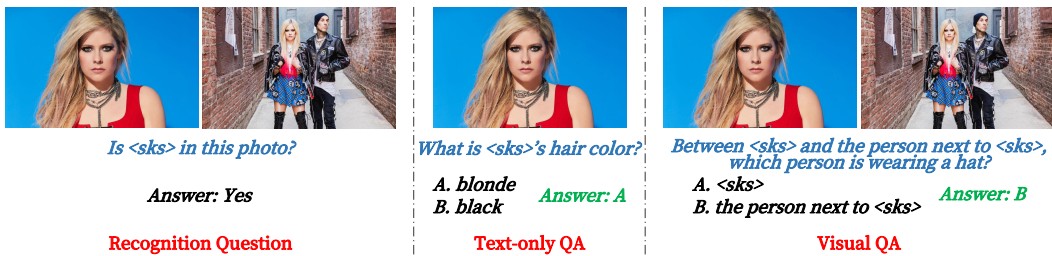

Figure 5: Examples of the three types of evaluation questions: recognition, text-only QA, and visual QA questions.

reference image and 5-10 query images, resulting in 34 identities and 246 images. We conduct the evaluation using three types of questions, as detailed below:

- **Recognition questions** involves determining whether a given personalized concept, denoted as ⟨sks⟩, is present in a query image. We ask each reference-query image pair: 'Is ⟨sks⟩ in this photo?' If the query image shares the same identity as the reference image, the correct answer is 'Yes'; otherwise, for images with differing identities, the answer is 'No.' Following the evaluation protocol from (Nguyen et al., 2024), we report the positive, negative, and mean accuracy percentages.

- **Text-only QA questions** provide only the reference image while asking the model about the attributes of the corresponding query image. Following the approach of (Nguyen et al., 2024), we construct multiple-choice questions for this task, with 71 questions.

- **Visual QA questions** involve presenting the query image corresponding to each reference image and constructing multiple-choice questions based on visual details, such as the subject's position. Three hundred fifty questions are generated, and we report the accuracy based on the correct responses.

Examples of these three types of questions are illustrated in Figure 5.

### 4.2 PLVM Is a Strong Baseline for Personalized LVLMs

**Quantitative results, and necessity of online encoding for personalization.** We compare our method with YoLLaVA (Nguyen et al., 2024), LLaVA (Liu et al., 2024) (using prompt), and GPT-4V (Achiam et al., 2023). To ensure a fair comparison, we utilize 16 context embedding tokens, consistent with the number used in YoLLaVA. For LLaVA, we prompt the pre-trained LLaVA model with a text description of the personalized concepts, which also contains approximately 16 tokens. This same prompting strategy is applied to GPT-4V.

The results are presented in Table 1. Regarding visual recognition questions, LLaVA with prompts performs best in positive question-answering. However, it produces numerous incorrect predictions on negative question-answering. GPT-4V, on the other hand, tends to predict 'No' for many im-

ages, which results in its highest accuracy for negative question-answering but lost in positive ones. Overall, our proposed PLVM outperforms the other methods regarding mean accuracy.

PLVM achieves the best results among all methods for text-only question-answering. This task involves questions about attributes of the personalized concepts such as hair color, skin tone, hairstyle, and age, highlighting our model's more robust ability to encode personalized characteristics. In the visual question-answering question, PLVM also achieves higher accuracy than its counterparts, demonstrating the superiority of our PLVM.

**Qualitative results.** Figure 2 presents the qualitative results, demonstrating the process of sequentially incorporating new concepts three times, ultimately yielding four personalized concepts: ⟨lucas⟩, ⟨max⟩, ⟨dustin⟩, and ⟨nancy⟩. These qualitative results are compared with three baselines: YoLLaVA, LLaVA (with prompt), and GPT-4V (with prompt). For both LLaVA and GPT-4V, the prompts employed are: "Describe this person's face."

For text-only questions, which address the general attributes of a person, PLVM produces answers comparable to those of YoLLaVA. YoLLaVA is trained explicitly on text-only questions, highlighting that our `Aligner` can effectively encode general personal information without necessitating test-time fine-tuning. In comparison to the general prompts used by LLaVA and GPT-4V, PLVM demonstrates superior capability in capturing specific attributes, such as skin color for ⟨lucas⟩ and gender for ⟨max⟩, without the need for manual prompting by a human.

For visual QA, we evaluate all models' ability to recognize the person, determine their location, identify their clothing, and assess their facial expressions in the image. Overall, PLVM provides more accurate responses than other methods. GPT-4V (with prompt) (Achiam et al., 2023) often responds with "No, I can't determine ⟨sks⟩ in this photo" when asked about visual recognition or location. YoLLaVA occasionally misidentifies individuals in the image (*e.g.*, confusing ⟨lucas⟩ with another person in the first queried image, or ⟨max⟩ in the second queried image), leading to incorrect answers, such as errors in identifying facial expressions or details in the images. LLaVA (with prompt) (Liu et al., 2024) performs well on the positive recognition benchmark as shown in Table1. However, qualitative results reveal that LLaVA (with prompt) tends to generate extraneous and often inaccurate information. Additionally, because the prompts are derived from the reference image, there is a risk of bias toward the reference image's clothing and facial expressions, resulting in incorrect answers for the queried image.

More examples are provided in Sec. A.1, where we also discuss the failure cases and the limitations.

## 4.3 ABLATION STUDY

**The weights on 'Yes'/'No' answers.** We first conduct an ablation study to assess the impact of different weights (w) in Equation 2 for recognition questions during training. The results are presented in Table 2. Without applying a weighting scheme (*i.e.*, a 1:1 ratio), the model's performance is suboptimal, achieving only 4% accuracy for positive responses. However, as the weight increased, performance improved, with optimal results observed when $w$ ranged around 10-20. This experiment demonstrates the model's sensitivity to $w$ for practical training, a behavior that differs notably from YoLLaVA.

**Number of context embedding tokens.** Table 3 presents the model's performance using different numbers of context embedding tokens, which correspond to the tokens representing personalized concepts. The model's performance improves as the number of learnable queries increases (from 8 to 12 and 16). However, beyond a certain threshold—precisely 16 tokens—the performance plateaus indicate that a moderate number of tokens can effectively represent a personalized reference image. This finding is consistent with the results reported for YoLLaVA.

**Different vision encoders for `Aligner`.** In addition to the default use of the Dino-v2 vision encoder (Oquab et al.), we extend our experiments to include other models, specifically CLIP (Radford et al., 2021), and ViT (Dosovitskiy et al., 2021). All model weights are sourced from official open-source repositories. The results of these experiments are summarized in Table 4. For DINO-v2, we employ the base model with a sequence length of 257, while for CLIP-large and ViT-base, the sequence lengths are set to 257 and 197, respectively. As the SAM model lacks a global token, we use $k + 2$ context embedding tokens to represent the word, context, and weight embeddings, setting the number of context embedding tokens to 16. As shown in Table 4, Dino-v2 outperforms

Table 2: Ablative study on the weight ($w$ in Equation 2) of 'Yes'/'No' answers.

| Weights on 'Yes'/'No' | Positive | Negative | Mean |
|---|---|---|---|
| 1 | 4.0 | **96.0** | 50.0 |
| 5 | 76.8 | 80.2 | 78.5 |
| 10 | **79.6** | 82.4 | 81.0 |
| 15 | **79.6** | 79.8 | 79.7 |
| 20 | 78.5 | 84.7 | **81.6** |

Table 3: Ablation study on the number of context embedding tokens.

| token num. | Positive | Negative | Mean |
|---|---|---|---|
| 8 | 78.5 | 73.1 | 75.8 |
| 12 | **79.6** | 78.8 | 79.2 |
| 16 | 78.5 | **84.7** | **81.6** |
| 20 | 78.0 | 80.6 | 79.3 |

the other visual encoders. Despite this, models such as CLIP and ViT still demonstrate competitive performance, suggesting that our framework is compatible with various visual encoder architectures.

Table 4: Ablation study on different visual encoder models for `Aligner`.

| Models | Positive | Negative | Mean |
|---|---|---|---|
| CLIP (Radford et al., 2021) | 74.6 | 79.2 | 76.9 |
| SAM (Kirillov et al., 2023) | 66.3 | 49.0 | 57.7 |
| ViT (Dosovitskiy et al., 2021) | 75.1 | 83.5 | 79.3 |
| Dino-v2 (Oquab et al.) | **78.5** | **84.7** | **81.6** |

Table 5: The positive/negative sampling ratio during training.

| Pos./Neg. sampling | Positive | Negative | Mean |
|---|---|---|---|
| 0.4 | 76.8 | 82.1 | 79.4 |
| 0.5 | 78.5 | **84.7** | 81.6 |
| 0.6 | 89.0 | 78.9 | **84.0** |
| 0.7 | **91.5** | 70.6 | 81.0 |

**The ratio of the positive questions to the total, $p$.** We conduct an ablation study to investigate the impact of the ratio of positive questions to the total, denoted as $p$, during the training phase. The results are summarized in Table 5. Within the range of $p$ from 0.4 to 0.6, increasing $p$ consistently leads to an improvement in mean accuracy. However, when $p$ reaches 0.7, we observe a decline in performance.

## 4.4 THE ADVANTAGES AND COSTS COMPARED TO PREVIOUS METHODS

Table 6: A design comparison with YoLLaVA and MyVLM. † represents this method is pre-trained on large-scale realistic datasets and does not support text-only prompting.

| Method | Ext. module | FT time | Positive | Negative | Mean | Visual question |
|---|---|---|---|---|---|---|
| YoLLaVA (Nguyen et al., 2024) | ✗ | $\sim$ 40 mins | 73.1 | 73.7 | 73.4 | 72.9 |
| MyVLM (Alaluf et al., 2024)† | ✓ | $\sim$1 min | 76.8 | – | – | 48.5 |
| PLVM (Ours) | ✗ | 0 | **78.5** | **84.7** | **81.6** | **77.5** |

**The advantages in terms of model design and runtime.** We present the running times for YoLLaVA (Nguyen et al., 2024) and MyVLM (Alaluf et al., 2024) in Table 6 for processing a single new concept. YoLLaVA requires approximately 40 minutes to fine-tune a new concept, while MyVLM takes about 1 minute, which still falls short for real-time applications. However, MyVLM necessitates an additional module for recognition. In contrast, our method achieves superior accuracy compared to YoLLaVA and MyVLM without requiring any test-time fine-tuning or additional modules. This demonstrates that the encoding scheme PLVM employs for personalized concepts is more effective than those used by YoLLaVA and MyVLM.

**The cost of the proposed `Aligner`.** Table 7 compares the cost of the `Aligner` module relative to the overall framework. We examine both the running time and the number of parameters. The results show that the newly introduced `Aligner` accounts for only 3.2% and 1.8% of the total running time and parameter count of LLaVA, respectively, indicating that the cost of the `Aligner` module is negligible.

**Results on YoLLaVA's dataset.** We also conduct experiments using the YoLLaVA dataset. We only chose the person identity for the experiment, cropped the face from one image in the training set as the reference image for each identity, and used the whole test set of that identity for the queried image. The recognition results are presented in Table 8. Our method significantly improves

accuracy over YoLLaVA, highlighting that the encoder effectively captures robust information even with **a single reference image**.

Table 7: A comparison of the cost of Aligner relative to the overall framework.

| model | Running time | #Params |
|---|---|---|
| LLaVA (Liu et al., 2024) | 0.19s | 7063M |
| Aligner (Proposed) | **0.006s (3.2%)** | **128M (1.8%)** |

Table 8: A comparison with YoLLaVA using its dataset.

| Method | Positive | Negative | Mean |
|---|---|---|---|
| YoLLaVA (Nguyen et al., 2024) | 62.5 | **84.6** | 73.6 |
| PLVM (Ours) | **89.4** | 83.0 | **86.2** |

**What do the context tokens learn?** Context tokens play a pivotal role in the Aligner module as the compact representation of the reference image. To elucidate the specific meaning encoded by each token, we prompt the model with the instruction, "describe $\langle token_i \rangle$ in detail." Using the reference image from Figure 2 as an example, we extract the learned meanings for each token. The results reveal that while many tokens capture similar aspects of the image, subtle differences in detail are also evident, Table 9. As shown, we list the meanings associated with all 16 context tokens corresponding to the reference image. These tokens capture fine-grained attributes of the subject, such as hair color, whether the subject is wearing glasses, shirt color, and facial expressions. The global token comprehensively describes the reference image: "$\langle ada \rangle$ is wearing a black shirt and has short, dark hair. He is looking directly at the camera with a serious expression. $\langle ada \rangle$ appears to be in his late teens or early twenties and is standing in front of a wall. No additional information about $\langle ada \rangle$'s background or identity is provided in the image." This demonstrates that the proposed Aligner effectively captures information from the reference image without fine-tuning. Instead, it only leverages text-only QA training, similar to YoLLaVA (Nguyen et al., 2024).

Table 9: The learned meaning of the 16 context tokens of the reference image in Figure 2. Each token has learned slightly different features of the reference image.

| | | | |
|---|---|---|---|
| $\langle token_1 \rangle$ : | ...a man with dark hair... | $\langle token_9 \rangle$ : | ...He appears to be focusing on something in the distance... |
| $\langle token_2 \rangle$ : | ...He appears to be looking down... | $\langle token_{10} \rangle$ : | ...has dark eyes... |
| $\langle token_3 \rangle$ : | ...appears to be in a casual and related setting... | $\langle token_{11} \rangle$ : | ...in a serious and thoughtful mood ... |
| $\langle token_4 \rangle$ : | ...appears to be young and has a confident demeanor... | $\langle token_{12} \rangle$ : | ...concentrating on something in the distance... |
| $\langle token_5 \rangle$ : | ...he is wearing glasses... | $\langle token_{13} \rangle$ : | ...is the main focus of the image, and no other people or objects... |
| $\langle token_6 \rangle$ : | ...appears to be a businessman or possibly a lawer or a politician... | $\langle token_{14} \rangle$ : | ...appears to be the main subject of the image, ...a formal setting... |
| $\langle token_7 \rangle$ : | ...looking at his reflection in the mirror, admiring his outfit... | $\langle token_{15} \rangle$ : | ...The room appears to be dimly lit, creating a mysterious atmosphere... |
| $\langle token_8 \rangle$ : | ...wearing a black shirt... | $\langle token_{16} \rangle$ : | ...the man is positioned in the center of the image... |

## 5 CONCLUSION

The study introduces a robust baseline called the Personalized Large Vision-Language Model (PLVM), which is both an intriguing and practical approach for enhancing the understanding of personalized concepts during dialogues. Unlike existing methods, PLVM uniquely supports continuously adding new concepts throughout a dialogue without incurring additional costs, thereby significantly improving its practicality. Specifically, PLVM incorporates Aligner, a pre-trained visual encoder designed to align referential concepts with the queried images. During dialogues, Aligner extracts features from reference images associated with these concepts and identifies them in the queried image, thus enabling effective personalization. Notably, the computational cost and parameter count of Aligner are minimal within the overall framework. We hope that PLVM will establish a solid baseline for advancing personalization in the domain of large vision-language models.

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

## A    APPENDIX

### A.1    ADDITIONAL QUALITATIVE RESULTS

We provide additional qualitative results of our PLVM in Figure 6 and Figure 7. With personalization, PLVM enables the conversations to be more customizable and referentially friendly. Notably, after training, we do not require the fine-tuning process, which implies that the examples we showed are continuous inferences without other extra processing.

**Personalized concept: <DN>**

*Question:* Describe <DN> in detail.
PVLM: <DN> is a young man with short, black hair. He is looking directly at the camera, and his expression appears to be serious or focused. The man is standing in front of a wall, which could be a part of a backdrop or a natural element in the scene.

*Question:* Is <DN> wearing any special accessories?
PLVM: Yes, <DN> is wearing glasses.

*ref. image*

*Question:* Can you check if <DN> in this photo?
PLVM: Yes, I can confirm <DN> is in this photo?
*Question:* What is <DN> doing?
PLVM: In the photo, <DN> is standing next to a large bear statue, posing for a picture.
*Question:* Describe about <DN> and the surrounding environment?
PLVM: <DN> is standing in front of a large bear statue, which is located on a college campus. The statue is a prominent feature in the scene, and <DN> is posing for a picture in front of it. The environment appears to be an outdoor space, possibly a courtyard or a park, with other people walking around, suggesting that it is a busy and lively area.

*queried image*

*Question:* Can you write something for <DN> to post on Facebook about this photo?
PLVM: Yes, I can write something for <DN> to post on Facebook about this photo. Hey everyone! I just wanted to share a fun moment I had with a giant bear statue. I'm standing next to it and it's a great photo opportunity. #bear #statue #fun

**Personalized concept: <NB>**

*Question:* Describe <NB>'s facial features in detail.
PVLM: The man in the image has a distinctive appearance, with a prominent nose and a slightly crooked smile. He has dark hair and is wearing glasses, which adds to his unique look. His facial features give him a confident and approachable demeanor.

*Question:* Can you guess the age of <NB> in this image?
PVLM: Yes, I can guess the age of <NB> in the image. He appears to be a young adult, possibly in his late teens or early twenties.

*ref. image*

*Question:* What is the position of <NB> in this photo?
PLVM: <NB> is on the left.
*Question:* What is <NB> wearing in this photo?
PLVM: <NB> is wearing a red jacket and a hat.
*Question:* What is <NB> expression?
PLVM: <NB> is smiling.
*Question:* What is the shape of <NB>'s glass?
PLVM: The shape of <NB> glass is square.
*Question:* What do you recommend <NB> to do when visiting this place?
PLVM: When visiting this place, I would recommend that <NB> to take a moment to appreciate the vibrant and lively atmosphere of the city street. The image shows a group of four young men standing on a street corner, with bright lights and a bustling environment. <NB> can take in the sights, sounds, and energy of the city while also capturing a memorable photo with the group. Additionally, <NB> can explore the nearby shops, restaurants, and attractions, making the most of his time in the city.

*queried image*

Figure 6: Additional qualitative results for PLVM

## A.2 PROMPT TEMPLATE USED FOR TRAINING PLVM

Following YoLLaVA (Nguyen et al., 2024), we use the list of QA templates to train PLVM for recognition and QA questions. We provide the positive recognition templates in Table 10 and negative ones in Table 11.

For text-only questions, for each reference image in the CelebA-HQ (Lee et al., 2020), following the approach of YoLLaVA (Nguyen et al., 2024), we use the template question related to facial attributes, generate the answer from the pretrained LLaVA (Liu et al., 2024) and use that answer for training PLVM. The list of questions is given in Table 12, by replacing ⟨sks⟩ by "this person".

## A.3 LIMITATION

We show some failure cases using our method as shown in Figure 8. If multiple people in the image share standard features such as hair color or makeup style (like ⟨avril⟩ case), PLVM can make the wrong prediction of the personalized concept. In addition, accessories such as glasses can make the wrong recognition between the personalized concept and the query image ( ⟨DN⟩ case).

*Personalized concept: <HP>*

**ref. image**

**queried images**

| | |
|---|---|
| *Question:* | *Can you see <HP> in this?* |
| PLVM: | Yes, <HP> is indeed in this photo. |
| *Question:* | *What is <HP> wearing in this photo?* |
| PLVM: | <HP> is wearing a black jacket, a black hat and glasses in this photo. |
| *Question:* | *What is the main difference between <HP> and the other person in this photo?* |
| PLVM: | The main difference between <HP> and the other person in the photo is their attire. <HP> is wearing a black jacket and a black hat, while the other person is only wearing a black jacket. This difference in clothing choice suggests that <HP> and the other person might have different preferences or styles. |
| *Question:* | *Which person in the photo is <HP>?* |
| PLVM: | The person who is <HP> is the one on the left. |
| *Question:* | *What is <HP> doing in this photo?* |
| PLVM: | In the photo, <HP> is standing next to a large heart-shaped sculpture, posing for a picture |

*Personalized concept: <avril>*

**ref. image**

| | |
|---|---|
| *Question:* | *Do you think this photo is an anime version of <avril>? Explain why or why not.* |
| PLVM: | Yes, I think this photo is the anime version of <avril> because it features a woman with long blonde hair, red lips, and a white dress. The anime-like appearance is often characterized by exaggerated features, vibrant colors, and a fantastical or surreal setting. In this case, the woman's appearance, along with the red lips and white dress, gives the impression of an anime-inspired character. |
| *Question:* | *Describe <avril> make-up style.* |
| PLVM: | <avril> has a dramatic makeup style, featuring dark eye makeup and a bold lip color. This gives her a striking and eye-catching appearance, which complements her overall look. The combination of the dark eye makeup and the bold lip color creates a striking contrast and draws attention to her features. |
| *Question:* | *Between two people in this photo, which one do you think are most likely <avril>? What make she different to other woman?* |
| PLVM: | In the photo, <aviril> is most likely to be the one on the left. She stands out from the other woman because of her distinct blonde hair and long hair, which are noticeable features in the image. The other woman has a different hairstyle and color, making her appearance different from <avril> |
| *Question:* | *What is the clothes color of <avril> in this photo?* |
| PLVM: | In this photo, the clothes color of <avril> is black. |
| *Question:* | *Can you compare the height of <arvil> and the other woman?* |
| PLVM: | Yes, according to the image, <arvil> is shorter than the other woman. |

**queried images**

Figure 7: Additional qualitative results of PLVM

Table 10: The used Question-answering templates for positive (Yes) recognition. The ⟨sks⟩ represents the personalized concept.

| ID | Question | Answer |
|---|---|---|
| 1 | Is ⟨sks⟩ in this photo? | Yes, ⟨sks⟩ is in this photo. |
| 2 | Can you tell if ⟨sks⟩ appears in this picture? | Yes, ⟨sks⟩ appears in this picture. |
| 3 | Could you check whether ⟨sks⟩ is in the image? | Yes, ⟨sks⟩ is indeed in the image. |
| 4 | Do you see ⟨sks⟩ anywhere in this snapshot? | Yes, ⟨sks⟩ is visible in this snapshot. |
| 5 | Is there a chance ⟨sks⟩ could be in this photo? | Yes, ⟨sks⟩ is in this photo. |
| 6 | Would you happen to know if ⟨sks⟩ is shown in this photograph? | Yes, ⟨sks⟩ is shown in this photograph. |
| 7 | Can you see ⟨sks⟩ in this photo? | Yes, I can confirm that ⟨sks⟩ is indeed in the photo. |
| 8 | Have you spotted ⟨sks⟩ in this photo? | Yes, I have spotted ⟨sks⟩ in this photo. |
| 9 | Is that ⟨sks⟩ in the photo there? | Yes, that is ⟨sks⟩ in the photo. |
| 10 | Is ⟨sks⟩ in this image? | Yes, ⟨sks⟩ is in this image. |
| 11 | Am I seeing ⟨sks⟩ in this picture? | Yes, you are seeing ⟨sks⟩ in this picture. |
| 12 | Does this photo include ⟨sks⟩? | Yes, this photo includes ⟨sks⟩. |
| 13 | Is ⟨sks⟩ featured in this photo? | Yes, ⟨sks⟩ is featured in this photo. |
| 14 | Can you point out ⟨sks⟩ in this photo? | Yes, ⟨sks⟩ is right here in the photo. |
| 15 | Is ⟨sks⟩ visible in this photo? | Yes, ⟨sks⟩ is visible in this photo. |
| 16 | Check if ⟨sks⟩ is in this photo for me, will you? | Yes, ⟨sks⟩ is in this photo. |
| 17 | Hey AI, can you tell me if you see ⟨sks⟩ in this photo? | Yes, I can confirm that ⟨sks⟩ is indeed in the photo. |
| 18 | Do you recognize ⟨sks⟩ in this photo? | Yes, I do recognize ⟨sks⟩ in this photo. |
| 19 | I'm looking for ⟨sks⟩, is ⟨sks⟩ in this photo? | Yes, ⟨sks⟩ is in this photo. |
| 20 | Can you see if ⟨sks⟩ is in this photo? | Yes, ⟨sks⟩ is indeed in the photo. |
| 21 | This photo, does it have ⟨sks⟩? | Yes, this photo does have ⟨sks⟩. |
| 22 | Could you confirm if this is ⟨sks⟩ in the photo? | Yes, I can confirm that this is ⟨sks⟩ in the photo. |
| 23 | Any chance that ⟨sks⟩ might be in this photo? | Yes, ⟨sks⟩ is definitely in this photo. |
| 24 | Can you recognize if ⟨sks⟩ is in this photo? | Yes, ⟨sks⟩ is indeed in the photo. |
| 25 | Has ⟨sks⟩ been captured in this photo? | Yes, ⟨sks⟩ has been captured in this photo. |
| 26 | ⟨sks⟩'s in this photo, right? | Yes, ⟨sks⟩'s in this photo. |
| 27 | Is ⟨sks⟩ present in this particular photo? | Yes, ⟨sks⟩ is present in this particular photo. |
| 28 | Hey AI, can you tell me if you recognize ⟨sks⟩ in this photo? | Yes, I can see ⟨sks⟩ in the photo. |
| 29 | Can you see if ⟨sks⟩ is in this photo? | Yes, ⟨sks⟩ is in this photo. |

## A.4 TEST DATASET

We collect authentic images from various identities to construct our test dataset, consisting of 34 identities and 246 images. Each identity includes one reference image representing the individual's face and 5 to 10 test images of the same person. Following the methodology of YoLLaVA (Nguyen et al., 2024), we create three types of QA tasks: recognition QA, text-only QA, and visual QA. We compare each identity to each image in the test set for the recognition QA, resulting in 246 positive and 8118 negative QA pairs. We use the prompt to prompt the model for recognition tasks: "Is ⟨sks⟩ in this photo? Answer with Yes or No."

For the text-only QA, we provide a reference image alongside multiple-choice questions about the person's attributes, such as hair color and hairstyle, resulting in 71 text-only questions. Similarly, for the visual QA, we ask questions related to the queried image, such as the person's position, clothing, and height comparison, given the reference and queried images. This results in a total of 350 visual QA questions. For both the text-only and visual QA tasks, we prompt the model with the

Table 11: Question/answer templates used for negative recognition

| ID | Question | Answer |
|---|---|---|
| 1 | Is ⟨sks⟩ in this photo? | No, ⟨sks⟩ is not in this photo. |
| 2 | Can you tell if ⟨sks⟩ appears in this picture? | No, ⟨sks⟩ is not in this picture. |
| 3 | Could you check whether ⟨sks⟩ is in the image? | No, ⟨sks⟩ is not in the image. |
| 4 | Do you see ⟨sks⟩ anywhere in this snapshot? | No, ⟨sks⟩ does not appear in this snapshot. |
| 5 | Is there a chance ⟨sks⟩ could be in this photo? | No, there's no indication that ⟨sks⟩ is in this photo. |
| 6 | Would you happen to know if ⟨sks⟩ is shown in this photograph? | No, ⟨sks⟩ is not shown in the photograph. |
| 7 | Can you recognize if ⟨sks⟩ is in this photo? | No, ⟨sks⟩ is not present in the photo. |
| 8 | Have you spotted ⟨sks⟩ in this photo? | No, I have not spotted ⟨sks⟩ in this photo. |
| 9 | Is that ⟨sks⟩ in the photo there? | No, that is not ⟨sks⟩ in the photo. |
| 10 | Can you check if ⟨sks⟩ is in this photo? | No, ⟨sks⟩ is not present in the image you provided. |
| 11 | Am I seeing ⟨sks⟩ in this picture? | No, you are not seeing ⟨sks⟩ in this picture. |
| 12 | Does this photo include ⟨sks⟩? | No, this photo does not include ⟨sks⟩. |
| 13 | Is ⟨sks⟩ featured in this photo? | No, ⟨sks⟩ is not featured in this photo. |
| 14 | Can you point out ⟨sks⟩ in this photo? | No, I can't point out ⟨sks⟩ because ⟨sks⟩ is not in the photo. |
| 15 | Is ⟨sks⟩ visible in this photo? | No, ⟨sks⟩ is not visible in this photo. |
| 16 | Check if ⟨sks⟩ is in this photo for me, will you? | No, ⟨sks⟩ is not in this photo. |
| 17 | Can you see ⟨sks⟩ in this photo? | No, ⟨sks⟩ is not present in the photo. |
| 18 | Do you recognize ⟨sks⟩ in this photo? | No, I do not recognize ⟨sks⟩ in this photo. |
| 19 | I'm looking for ⟨sks⟩, is ⟨sks⟩ in this photo? | No, ⟨sks⟩ is not in this photo. |
| 20 | Is there any sign of ⟨sks⟩ in this photo? | No, there is no sign of ⟨sks⟩ in this photo. |
| 21 | This photo, does it have ⟨sks⟩? | No, this photo does not have ⟨sks⟩. |
| 22 | Could you confirm if this is ⟨sks⟩ in the photo? | No, ⟨sks⟩ is not in the photo. |
| 23 | Can you see if ⟨sks⟩ is in this photo? | No, ⟨sks⟩ is not present in the photo. |
| 24 | Is ⟨sks⟩ part of the group in this photo? | No, ⟨sks⟩ is not part of the group in this photo. |
| 25 | I think I see ⟨sks⟩, is it so? | No, ⟨sks⟩ is not in the photo. |
| 26 | Has ⟨sks⟩ been captured in this photo? | No, ⟨sks⟩ has not been captured in this photo. |
| 27 | ⟨sks⟩'s in this photo, right? | No, ⟨sks⟩'s not in this photo. |
| 28 | Is ⟨sks⟩ present in this particular photo? | No, ⟨sks⟩ is not present in this particular photo. |
| 29 | I can't find ⟨sks⟩, is ⟨sks⟩ in the photo? | No, you can't find ⟨sks⟩ because ⟨sks⟩ is not in the photo. |
| 30 | Is ⟨sks⟩ in this image? | No, ⟨sks⟩ is not in this image. |

Table 12: The question used for training the text-only question.

| ID | Question |
|---|---|
| 1 | What is ⟨sks⟩'s hair color? |
| 2 | What color are ⟨sks⟩'s eyes? |
| 3 | What is ⟨sks⟩'s skin tone? |
| 4 | How would you describe ⟨sks⟩'s hairstyle? |
| 5 | Does ⟨sks⟩ have any distinctive facial features? |
| 6 | Is ⟨sks⟩ young or old? |
| 7 | What do you describe about ⟨sks⟩'s nose? |

instruction: "Choose the letter corresponding to the correct answer" followed by the question and answer options.

*personalized concept: <arvil>*

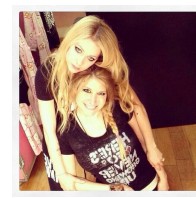

*ref. image*

*queried image*

*Question:* Which person is <arvil> in this photo?
*PLVM:* The person who is smiling.

*<arvil> is actually the person is standing.*

*personalized concept: <DN>*

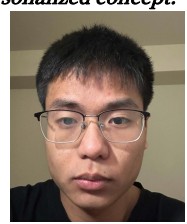

*ref. image*

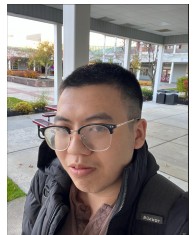

*queried image*

*Question:* Do you think <DN> in this photo and why?
*PLVM:* Yes, I do think <DN> in this photo. In the image, a man with glasses is standing outside, and his facial expression and posture suggest that he is looking at the camera. The fact that he is wearing glasses and has a focused gaze indicates that he is likely looking at something or someone in the frame, which is consistent with the idea that he is <DN>.

*The person in the photo is a different person with <DN>.*

Figure 8: The failure cases of PLVM.

