# OpenReview forum: "Personalized Large Vision-Language Model"
_ICLR.cc/2025/Conference — ICLR 2025 Conference Withdrawn Submission_

### Official Review · Reviewer_EYaB · 2024-10-21

**Soundness:** 3
**Presentation:** 4
**Contribution:** 2
**Rating:** 5
**Confidence:** 4

**Summary:**

The paper presents a personalization LVLM capable of interactive dialogues using referential concepts. The authors introduces an additional vision encoder for concept embedding. A training dataset is curated by using IP-Adapter to generate various images of humans. The fine-tuning process enables the model to handle new concepts without training. Both qualitative and quantitative results are provided to validate the effectiveness of the proposed method.

**Strengths:**

+ The proposed method consistently outperforms previous methods.
+ No inference cost is introduced for new concepts.

**Weaknesses:**

+ LVLMs trained with multi-images like [1] and [2] should also be evaluated. Since the proposed method actually introduces an additional vision encoder for the reference image, it would be beneficial to show whether using the original vision encoder of these LVLM can complete the task well. If current models can effectively handle these tasks using their built-in encoders, this would significantly impact the necessity and contribution of the proposed additional encoder architecture.

+ The proposed method seems to be limited to only identifying humans, while [3] can address both humans and other creatures like dogs or toys. It seems to me that the method trained a vision encoder specifically for facial recognition, and other objects cannot be handled. The generalization ability of the method is restricted.

+ The comparisons in Fig. 4 and Table 1 are unfair. Both LLaVA and GPT-4V only receive textual prompts of the concept. However, PLVM receives the reference image. Obviously, the reference image contains much more information than its textual prompt. The content of a human face is difficult to fully describe only through language. The input information is not set to equal so the comparison is meaningless.




[1] Li, Bo, et al. "Llava-onevision: Easy visual task transfer." arXiv preprint arXiv:2408.03326 (2024).

[2] Wang, Peng, et al. "Qwen2-VL: Enhancing Vision-Language Model's Perception of the World at Any Resolution." arXiv preprint arXiv:2409.12191 (2024).

[3] Nguyen, Thao, et al. "Yo'LLaVA: Your Personalized Language and Vision Assistant." arXiv preprint arXiv:2406.09400 (2024).

**Questions:**

+ What is the scope of the aligner module beyond human subjects? For example, can it handle concepts like a cat, a house, or a toy?
+ Can the model handle multiple concepts in a single prompt? For example, if person A, B and C are all present in the photo, and the reference images are person A and B, can it answer: 'Is A on the left or right of B?'

---

### Official Review · Reviewer_Nfjz · 2024-11-01

**Soundness:** 2
**Presentation:** 3
**Contribution:** 2
**Rating:** 5
**Confidence:** 5

**Summary:**

- **Personalization in LVLMs**: The paper introduces Personalized Large Vision-Language Models (PLVMs) that enhance user interaction by recognizing and incorporating personalized concepts in dialogues.
- **Aligner Method**: PLVM uses a pre-trained visual encoder called Aligner to embed personalized concepts online, enabling efficient recognition without additional fine-tuning.
- **Practical Applications**: The model is designed for practical use in dialogue systems, question-answering, and human-computer interaction, making conversations more intuitive and customizable.
- **Experimental Validation**: Comprehensive qualitative and quantitative analyses demonstrate the effectiveness and superiority of PLVM compared to existing methods.

**Strengths:**

The paper introduces an approach to personalizing large vision-language models (LVLMs) by embedding personalized concepts online using a pre-trained visual encoder called Aligner. This method stands out due to its ability to recognize and incorporate personalized concepts without additional fine-tuning, which is a significant departure from traditional methods that often require extensive retraining. The originality lies in the creative combination of existing ideas in visual encoding and personalization, applied to enhance user interaction in dialogue systems.

Overall, the paper makes a valuable contribution to the field of vision-language models, offering a novel and practical solution to personalization that is both efficient and effective. The combination of originality, quality, clarity, and significance makes this work a noteworthy addition to the literature.

**Weaknesses:**

This topic is not new but authors just try it again by constructing another dataset. As a result, authors should stress more on the importance and challenges as well as some new insights for this task.

1. Clarification on <sks> token: Why is the personalized concept denoted as <sks>? This isn’t a major issue, but I’m curious about the reasoning behind this choice.
2. Expand Experimental Section: The experiments primarily focus on synthetic datasets and a limited number of real-world images. Please expand the experimental section to include more diverse and real-world datasets. Conduct additional experiments to validate the model’s performance in various practical scenarios, such as different lighting conditions, occlusions, and varying image resolutions.
3. Clarify LLaVA and LLaVA (with prompt): What is the difference between LLaVA and LLaVA (with prompt)? How do they work? Please state this clearly.
4. Ensure Fair Comparisons: How is the data quality of the test set ensured? GPT-4V is a strong baseline. Since the state-of-the-art methods in Table 1 are trained on different domains, the authors should add a baseline with LLaVA.
5. Impact of MLPs and Cross-Attention: How do MLPs and cross-attention affect performance? Why can’t all visual inputs share one encoder? The authors should remove these modules to validate their efficacy.

**Questions:**

Please see Weaknesses.

---

### Official Review · Reviewer_mry6 · 2024-11-04

**Soundness:** 2
**Presentation:** 3
**Contribution:** 1
**Rating:** 3
**Confidence:** 4

**Summary:**

The paper introduces Personalized Large Vision-Language Models (PLVM), which enhance the ability of vision-language models (LVLMs) to recognize and interact with personalized concepts without needing extensive fine-tuning. PLVM uses an "Aligner," a pre-trained visual encoder, to embed concepts online, enabling seamless integration of new concepts and making personalized interactions more efficient. This approach significantly reduces computational costs and allows for real-time application, unlike previous methods that require lengthy adjustments for personalization.

The authors demonstrate that PLVM outperforms existing models in tasks such as recognition, text-only question-answering, and visual question-answering. The model's design allows it to adapt to new personalized concepts without incurring additional costs, showcasing its practicality for use in dialogue systems and human-computer interactions. Through comprehensive experiments and comparisons, the paper highlights PLVM's effectiveness and positions it as a strong baseline for personalized LVLMs.

**Strengths:**

- PLVM introduces the Aligner method, enabling real-time personalization without test-time fine-tuning, unlike models like YoLLaVA.
- Demonstrates superior performance in tasks like visual recognition and question-answering, with higher mean accuracy compared to other models.
- Ablation studies in Table 2 and Table 3 demonstrate the effect of varying parameters like context embedding tokens and weights for ‘Yes’/‘No’ answers, reinforcing the robustness of the model’s design.
- The approach is highly practical, allowing seamless integration of new personalized concepts without additional costs or re-training.

**Weaknesses:**

- The paper briefly touches on the comparative performance of different visual encoders (e.g., CLIP, ViT) but lacks detailed analysis on how these affect accuracy, computational cost, and adaptability; expanding this section would be beneficial.
- The experiments primarily address simple visual and recognition tasks, without exploring complex questions involving object relationships or action-based elements.
- The evaluation, using 246 images with 34 identities, may not reflect scalability; insights on performance with larger, more diverse datasets would strengthen the paper.
- PLVM's training on a tailored dataset gives it an advantage over models that lack this specific training; fine-tuning competing models on similar data or providing their expected performance under the same conditions would make comparisons fairer.

**Questions:**

- Can you provide more details on how different visual encoders (e.g., CLIP, ViT) impact PLVM's performance in terms of accuracy, computational cost, and adaptability to various use cases? This would help assess the flexibility and potential trade-offs of using alternative encoders.
- How well does PLVM generalize to real-world, unstructured data beyond the synthesized dataset? Including results or insights from tests on more diverse and complex datasets would clarify the model's applicability in broader contexts.
- What measures were taken to ensure that the synthesized dataset used for training does not introduce biases that might skew PLVM's performance? An explanation of the dataset's diversity and representativeness would be helpful.

---

### Official Review · Reviewer_kLHY · 2024-11-04

**Soundness:** 2
**Presentation:** 2
**Contribution:** 1
**Rating:** 3
**Confidence:** 5

**Summary:**

The paper introduces the Personalized Large Vision-Language Model (PLVM), which enhances the interaction with large vision-language models by integrating personalized concepts without requiring extensive fine-tuning. Central to this approach is the "Aligner," a pre-trained visual encoder that dynamically encodes personalized references in real-time. The methodology involves generating a synthetic dataset of paired images, which are used to train the model through a weighted loss function that emphasizes accurate recognition. Experimental evaluations demonstrate PLVM's superior performance in visual recognition and question-answering tasks compared to existing models like YoLLaVA and LLaVA, highlighting its efficiency in incorporating new concepts seamlessly.

**Strengths:**

1. The introduction of the Aligner module allows for real-time personalization without the need for test-time fine-tuning.
2. The creation of a comprehensive synthetic dataset enhances the model's ability to recognize diverse personalized concepts.

**Weaknesses:**

1. Figure 2 needs to be redrawn; it seems to have little content but occupies a large portion of the space.
2. The performance in Figure 4 seems to show little difference. The authors need to clearly express the intended message.
3. The paper lacks a comparison with the RAG method.
4. Given that training-free methods like RAG have become popular solutions, the paper proposes a training method. The authors need to emphasize the contribution and significance of this approach.
5. The motivation and figures in the paper seem disorganized and require further refinement.

**Questions:**

1. The proposed model in the paper requires training. What advantages does it have compared to RAG?
2. Could you provide a comparison with the RAG method?

---

### Note · Authors · 2024-11-15

I have read and agree with the venue's withdrawal policy on behalf of myself and my co-authors.